# Influence of CYP2D6, CYP3A, and ABCG2 Genetic Polymorphisms on Ibrutinib Disposition in Chinese Healthy Subjects

**DOI:** 10.3390/ph18111615

**Published:** 2025-10-26

**Authors:** Kejia Fu, Yao Wang, Lingyan Duan, Zhenyuan Zhang, Jialing Qian, Xijing Chen, Yi Liang, Chengcan Lu, Di Zhao

**Affiliations:** 1Clinical Pharmacology Research Center, School of Basic Medicine and Clinical Pharmacy, China Pharmaceutical University, Nanjing 211198, China; 2School of International Pharmaceutical Business, China Pharmaceutical University, Nanjing 211198, China; 3Department of Pharmacy, The Affiliated Jiangning Hospital of Nanjing Medical University, Nanjing 211100, China; 4Post-Doctoral Research Center, Nanjing First Hospital, Nanjing Medical University, Nanjing 210006, China

**Keywords:** ibrutinib, polymorphism, pharmacokinetics, CYP3A, ABCG2

## Abstract

**Objectives**: This study aimed to elucidate the determinants of interindividual variability in the pharmacokinetics of ibrutinib among healthy Chinese subjects, focusing on the influence of demographic characteristics, dietary conditions, and genetic polymorphisms on CYP enzymes and ABC transporters. **Methods**: Thirty-two participants were randomly assigned to either a fasting (n = 16) or fed (n = 16) group, each receiving a single 140 mg oral dose of ibrutinib. Plasma concentrations were quantified using a validated UPLC–MS/MS method. Genetic polymorphisms in CYP3A4, CYP3A5, CYP2D6, and ABCG2 were identified by Sanger sequencing. Pharmacokinetic parameters, including apparent clearance (CL/F), maximum plasma concentration (Cmax), area under the plasma concentration–time curve (AUC0-t), and time to maximum concentration (Tmax), were estimated by non-compartmental analysis and statistically evaluated for associations with demographic, dietary, and genetic variables. **Results**: Food intake significantly affected ibrutinib pharmacokinetics, with postprandial administration resulting in reduced CL/F and increased Cmax and AUC0-t (*p* < 0.01). Gender differences were also observed, as females exhibited higher CL/F, lower Cmax, and AUC0-t than males (*p* < 0.05). The CYP2D6 c.100C>T polymorphism significantly decreased CL/F and increased exposure in fasting and male subjects (*p* < 0.05), but this effect was absent under fed conditions. Conversely, the ABCG2 c.421C>A variant was associated with increased CL/F and decreased AUC0-t (*p* < 0.05), while other genotypes exerted negligible effects. **Conclusions**: Ibrutinib pharmacokinetics are significantly modulated by dietary status, gender, and genetic polymorphisms, particularly CYP2D6 c.100C>T and ABCG2 c.421C>A. These findings underscore the importance of integrating pharmacogenetic and physiological factors into individualized dosing strategies to optimize therapeutic efficacy and minimize adverse effects.

## 1. Introduction

Ibrutinib, an oral Bruton’s tyrosine kinase (BTK) inhibitor, was approved by the FDA in 2013 and by the NMPA in 2017 for the treatment of patients with chronic lymphocytic leukemia (CLL) or small lymphocytic lymphoma (SLL) who have received at least one prior therapy [1,2]. BTK, a member of the TEC family of non-receptor tyrosine kinases, plays a central role in B-cell receptor signaling, regulating B-cell proliferation, differentiation, and apoptosis [3,4,5]. By covalently binding to a cysteine residue within the BTK active site (IC50 = 0.5 nmol/L), ibrutinib irreversibly inhibits BTK activity, thereby suppressing malignant B-cell proliferation and migration to tumor-supportive lymphoid microenvironments [6,7].

Ibrutinib exhibits low bioavailability and high interindividual pharmacokinetic variability due to extensive first-pass metabolism by cytochrome P450 (CYP) enzymes [8]. Both in vitro and in vivo evidence identify CYP3A as the primary enzyme involved, with additional contribution from CYP2D6 [9]. Moreover, ibrutinib is a substrate of the ATP-binding cassette (ABC) transporter ABCG2 [10]. Over the past decades, extensive research has investigated the genetic polymorphisms of drug-metabolizing enzymes and transporters. For example, CYP3A4*1G and CYP3A5*3 alleles are prevalent in the Chinese population, with allele frequencies of 0.221 and 0.724, respectively [11,12,13]. In addition, CYP2D6*10 (c.100C>T) and ABCG2 (c.421C>A, c.34G>A) variants are prevalent in Asian populations, particularly among Chinese individuals, with a significant impact on drug exposure and pharmacokinetics [14,15,16,17]. Although current evidence indicates that polymorphisms in CYP3A4/5, CYP2D6, and ABCG2 may contribute to interindividual variability in pharmacokinetics, their effects on the in vivo disposition of ibrutinib remain insufficiently characterized. Apart from genetic variations, clinical and demographic characteristics may also contribute to pharmacokinetic variability. Previous studies have shown that dietary conditions can nearly double ibrutinib’s bioavailability, while age (17–84 years) and gender also have no clinically relevant impact on its pharmacokinetics [9]. Renal impairment has no significant effect on ibrutinib exposure, whereas hepatic impairment increases ibrutinib exposure, with its usage recommendation limited to patients with mild liver impairment and contraindicated in those with moderate or severe liver dysfunction [18]. Despite these insights, the effects of genetic and non-genetic factors on ibrutinib pharmacokinetics have not been systematically investigated.

This study aimed to investigate the influence of dietary intake, genetic variations in CYP-mediated metabolic enzymes and transporters, and demographic factors, including body mass index (BMI), serum alanine aminotransferase (ALT), serum aspartate aminotransferase (AST), serum creatinine (CREA), age, and gender, on the pharmacokinetic profile of ibrutinib. Our findings are expected to support the development of individualized dosing strategies for ibrutinib therapy.

## 2. Results

### 2.1. Study Population

The baseline demographics and clinical characteristics of the 32 healthy Chinese subjects enrolled are summarized in Table 1. The seven single-nucleotide polymorphisms (SNPs) analyzed, located in genes encoding CYP3A metabolic enzymes and ABC transporters, are presented in Table 2. The minor allele frequencies (MAFs) of these SNPs were consistent with those reported in East Asian populations from the 1000 Genomes Project database (http://www.1000genomes.org/). In addition, all SNPs were in Hardy–Weinberg equilibrium (HWE, *p* > 0.05), indicating that the genotype distributions of the study participants were representative of the general population.

### 2.2. Demographics and Dietary Impact on Ibrutinib Pharmacokinetics

Pearson correlation analysis showed no clinically meaningful associations between demographic variables (age, BMI, CREA, ALT, AST, body weight, and height) and the primary pharmacokinetic parameters of ibrutinib. The mean plasma concentration–time profiles and pharmacokinetic parameters of the 32 subjects following a single 140 mg dose of ibrutinib are presented in Figure 1 and Table 3.

Interestingly, sex-based differences were observed: apparent clearance (CL/F) in female subjects was significantly higher than in males, whereas both maximum plasma concentration (Cmax) and area under the plasma concentration–time curve (AUC0-t) values in females were approximately two-thirds of those in males (Figure 2a–d, *p* < 0.05). In addition, dietary intake also exerted a marked effect on ibrutinib pharmacokinetics. The Mann–Whitney U test demonstrated that postprandial administration significantly reduced CL/F (*p* < 0.01) while increasing Cmax and AUC0-t (*p* < 0.01) and prolonging time to maximum concentration (Tmax, *p* < 0.01) (Figure 2e–h). Overall, the geometric mean exposure in the fed state was approximately 1.7-fold higher than that in the fasting state.

### 2.3. Effects of CYP2D6, CYP3A4/5, and ABCG2 Polymorphisms on Ibrutinib Pharmacokinetics

Mean plasma concentration–time profiles and pharmacokinetic parameters between different genotype groups are presented in Figure 3 and Table 4, corresponding to SNPs in CYP2D6 (c.100C>T, c.2851 C>T), CYP3A4 c.22545 G>A, CYP3A5 c.6986 G>A, and ABCG2 (c.34G>A, c.421C>A).

The Mann–Whitney U test (Figure 4) revealed that the CYP2D6 c.100C>T mutation significantly reduced CL/F (*p* = 0.001) and increased Cmax and AUC0-t (*p* < 0.05). After adjustment for gender and food intake to minimize confounding, subgroup analysis showed that the CYP2D6 c.100C>T mutation resulted in decreased CL/F and increased AUC0-t in male subjects (Figure 5a–d). Among fasted subjects, the mutation was associated with a reduction in CL/F and increases in Cmax and AUC0-t (Figure 5e–h). However, no significant differences were observed between wild-type and mutant subjects in the postprandial state (Figure 5i–l).

At the CYP3A4 c.22545G>A locus, heterozygous subjects (GA) exhibited a significant reduction in CL/F (*p* = 0.040), with a non-significant trend toward increased Cmax and AUC0-t (Table 4). For the ABCG2 c.421C>A locus, heterozygous carriers (CA) demonstrated significantly higher CL/F and lower AUC0-t compared with wild-type subjects (CC) (Figure 6). No statistically significant effects on the pharmacokinetics of ibrutinib were observed for mutations at the CYP2D6 c.2851 C>T, CYP3A5 c.6986 G>A, or ABCG2 c.34G>A loci (Table 4).

## 3. Discussion

This study comprehensively evaluated the factors contributing to interindividual variability in the pharmacokinetics of ibrutinib capsules among healthy Chinese subjects, focusing on demographic characteristics, dietary conditions, and genetic polymorphisms in CYP enzymes and ABC transporters. No significant correlations were observed between age, BMI, body weight, or height and the key pharmacokinetic parameters of ibrutinib. Although systemic exposure was approximately 1.4-fold higher in males than in females, previous studies have indicated that this difference is not clinically meaningful with respect to pharmacokinetics [9,19]. Additionally, no significant associations were observed between ALT, AST, or CREA levels and the pharmacokinetics of ibrutinib, which is consistent with previous findings, indicating a minimal renal impairment effect on ibrutinib exposure and its safe use in patients with mild hepatic impairment [19,20].

Food intake exerted a substantial influence on ibrutinib pharmacokinetics, as reflected by delayed gastric emptying and prolonged time to peak concentration. In our cohort, systemic exposure of ibrutinib increased by approximately 1.7-fold under fed conditions compared with fasting, consistent with previous reports. According to the FDA prescribing information, coadministration with a high-fat, high-calorie meal (800–1000 kcal with ~50% of calories from fat) increases Cmax by 2–4 fold and AUC by roughly twofold [9]. Similarly, de Jong et al. observed a 2–3-fold increase in Cmax and nearly doubled AUC in non-Chinese adults, while a recent randomized crossover study in healthy Chinese volunteers reported 2.2-fold and 1.3-fold elevations in Cmax and AUC, respectively [20,21]. Collectively, these findings indicate that the magnitude of the food effect observed in our study is within the expected range across ethnic groups and study designs. Several physiological mechanisms may account for this enhanced exposure in the fed state, including delayed gastric emptying, altered gastrointestinal pH, and increased splanchnic blood flow [20]. Ibrutinib is classified as a Biopharmaceutics Classification System (BCS) class II compound, characterized by high permeability but low aqueous solubility [22]. Regulatory data indicate that its solubility is pH-dependent and remains practically insoluble across the physiological pH range of 3–8 [23]. Despite the decrease in solubility due to elevated gastrointestinal pH during feeding, the study observed that exposure levels increased and CL/F rates declined postprandially (*p* < 0.01, Figure 5). These findings may be attributed to increased intestinal blood flow, which enhances ibrutinib absorption into the portal circulation and mitigates the first-pass effect from intestinal CYP enzymes. The apparent volume of distribution (Vz/F) decreased from 18,727 L (fasted) to 3984 L (fed), further supporting this hypothesis (Appendix A).

Although CYP2D6 is not recognized as a major enzyme for ibrutinib metabolism, the CYP2D6 c.100C>T polymorphism was found to significantly decrease CL/F and increase drug exposure, particularly among fasting and male subjects. However, this effect was not observed under postprandial conditions, suggesting that food intake may override the genetic influence of CYP2D6 on ibrutinib metabolism. Previous studies have indicated that CYP3A genetic polymorphisms rarely produce clinically meaningful variability in ibrutinib pharmacokinetics, which aligns with our findings showing minimal effects of CYP3A4/5 variants in this cohort [24]. Interestingly, we observed that the ABCG2 c.421C>A mutation was associated with decreased ibrutinib exposure, which contrasts with the common expectation that ABCG2 variants increase systemic exposure to substrates [13,25]. However, this finding may be influenced by the limited sample size and the exploratory nature of the analysis without adjustment for multiplicity. Therefore, the result should be interpreted with caution, and larger studies with appropriate statistical control are needed to verify and elucidate this observation.

In addition, growing evidence indicates that interethnic differences in pharmacogenetic backgrounds may influence ibrutinib disposition. Data from the 1000 Genomes Project show that the CYP2D6 c.100C>T variant has a minor allele frequency of ~0.57 in East Asians, compared with 0.15–0.20 in Americans and Europeans. Similarly, the ABCG2 c.421C>A allele is more common in East Asians (0.29) than in Americans (0.14) or Europeans (0.09) (Table 2). These disparities suggest that the genotype–pharmacokinetic associations observed in Chinese individuals may not be directly extrapolated to other ethnic populations. Prior studies have also reported that reduced-function CYP2D6 alleles are associated with increased systemic exposure and adverse effects, and that ABCG2 c.421A may alter efflux activity, drug bioavailability, and toxicity risk—findings that align with our data [26]. Moreover, a recent retrospective pharmacogenetic study showed that high-risk genotypes in KCNQ1 and GATA4, together with variants in CYP2D6, CYP3A4, CYP3A5, and ABCB1, were linked to increased cardiovascular adverse events and treatment modifications during ibrutinib therapy [27]. Although that study addressed toxicity rather than pharmacokinetics, it underscores the broader clinical implications of genetic variability. Collectively, these observations highlight the importance of validating genotype–PK associations in diverse populations and considering ethnic differences when developing precision dosing strategies.

Taken together, this study provides evidence that the CYP2D6 c.100C>T and ABCG2 c.421C>A polymorphisms have differential effects on the pharmacokinetics of ibrutinib, whereas other key genes involved in drug metabolism and transport cannot be excluded. For instance, previous work from our group showed that the ABCB1 3435C>T polymorphism significantly altered the exposure and CL/F of Lenvatinib [16]. In addition, the ABCC2 c.-24C>T variant was associated with the pharmacokinetic variability of deferasirox in Chinese populations [17]. All these findings, together with research output from other relevant studies, highlight the critical roles of polymorphisms in CYP enzymes and ABC transporters in modulating drug exposure and pharmacokinetic properties. However, several methodological limitations should be taken into account. First, no haplotype-based or phenotype-based classification was prespecified, and all genetic analyses were exploratory. Second, multiple comparisons were performed without adjustment for multiplicity, increasing the likelihood of type I error. Third, neither effect sizes nor 95% confidence intervals were calculated, and no predefined statistical analysis plan was implemented. Consequently, the present results should be interpreted with caution. Future studies with larger cohorts and prospective pharmacogenetic designs are needed to validate these associations and to construct genotype-informed models. Moreover, confirmation in patient populations under therapeutic dosing conditions will be essential to establish the clinical relevance of these preliminary observations.

In conclusion, this study demonstrates that food intake, gender, and genetic polymorphisms (CYP2D6 c.100C>T and ABCG2 c.421C>A) substantially contribute to interindividual variability in ibrutinib pharmacokinetics among healthy Chinese subjects. Given the exploratory nature of the analyses and the limited sample size, further prospective crossover studies or population pharmacokinetic modeling are warranted to validate these observations and clarify their implications for individualized ibrutinib dosing in clinical practice.

## 4. Materials and Methods

### 4.1. Study Design

A total of 32 healthy Chinese volunteers participated in a two-period, two-sequence crossover study. Volunteers were randomly assigned to one of two sequences: fasting–fed (n = 16) or fed–fasting (n = 16). Each participant received a single oral dose of 140 mg ibrutinib capsules (Imbruvica^®^, Pharmacyclics LLC, Sunnyvale, CA, USA, Batch Number: 1921112) under both fasting and fed conditions, with a washout interval of seven days between the two phases. In the fed phase, a standardized high-fat, high-calorie meal (~800–1000 kcal; approximately 50% of total calories from fat, 35% from carbohydrates, and 15% from protein) was consumed 30 min before dosing, in accordance with regulatory bioequivalence study guidance.

Venous blood samples (4 mL each) were collected into EDTA-K2 anticoagulant tubes at predose and at 0.25, 0.5, 0.75, 1, 1.33, 1.67, 2, 2.5, 3, 4, 6, 8, 10, 12, 24, 36, 48, and 60 h post-administration. Samples were immediately centrifuged at 3000× *g* for 5 min at 4 °C, and plasma aliquots were stored at −70 °C until pharmacokinetic analysis, while hemocytes were stored at −70 °C for subsequent genotyping analysis.

### 4.2. Determination of Ibrutinib in Human Plasma

Plasma concentrations of ibrutinib were determined using a validated ultra-performance liquid chromatography–tandem mass spectrometry (UPLC–MS/MS) method developed in our laboratory, following the ICH M10 (2022) guideline. The analytical system consisted of an ACQUITY UPLC H-Class system (Waters, Milford, MA, USA) coupled to a Triple Quad™ 6500+ Mass Spectrometer (AB SCIEX, Framingham, MA, USA) equipped with an electrospray ionization (ESI) source operating in positive mode.

Plasma proteins were precipitated by adding acetonitrile at a volume ratio of 4:1 (acetonitrile: plasma), followed by centrifugation. Chromatographic separation was achieved on an ACQUITY UPLC HSS^®^ T3 column under isocratic elution with a mobile phase consisting of acetonitrile and water containing 5 mM ammonium acetate (90:10, *v*/*v*). The flow rate was maintained at 0.4 mL/min, and the total run time was 2.5 min.

Quantitation was performed using multiple reaction monitoring (MRM) with transitions of *m*/*z* 441.3→138.1 for ibrutinib and *m*/*z* 446.3→309.1 for the internal standard (ibrutinib-d_5_). The method demonstrated good selectivity and linearity (0.5–500 ng/mL, r^2^ > 0.99), with acceptable accuracy and precision within ±15%. No significant matrix effect or carryover was observed, and the analyte was stable under all tested conditions. Representative chromatograms are shown in Appendix A. All plasma samples were handled using uniform automated procedures independent of grouping or genotyping information.

### 4.3. Genotyping Analysis

Genomic DNA was isolated from peripheral whole blood using a commercial extraction kit (Tiangen Biotech Co., Beijing, China) in accordance with the manufacturer’s instructions. Polymerase chain reaction (PCR) was employed to amplify seven target loci, including CYP3A4*1G (rs2242480), CYP3A5*3 (rs776746), CYP2D6 c.100C>T (rs1065852), CYP2D6 c.2851C>T (rs16947), ABCG2 c.421C>A (rs2231142), and ABCG2 c.34G>A (rs2231137).

The forward and reverse primer sequences are presented in Table 5. Genotyping analysis was performed by Sanger sequencing using an ABI 3730xl DNA Analyzer (Applied Biosystems, Foster City, CA, USA), and sequence data were analyzed with standard software packages (Informax Vector NTI Suite version 9.0).

### 4.4. Pharmacokinetic and Statistical Analysis

Key pharmacokinetic parameters, including Cmax, Tmax, AUC0-t, and CL/F, were calculated using non-compartmental analysis in Phoenix WinNonlin version 8.2 (Certara Inc., NJ, USA). Additional parameters, including AUC0-∞, λz, t_1_/_2_, Vz/F, and %AUC_extrapolated, were also derived and are presented in Appendix A.

All statistical analyses were performed using SPSS version 24.0 (IBM Corp., Armonk, NY, USA). Data were summarized as medians with interquartile ranges (IQRs). The Hardy–Weinberg equilibrium for each SNP was assessed using the Chi-squared test. Differences in pharmacokinetic parameters among genotypes were evaluated using the Mann–Whitney U test or the Kruskal–Wallis H test, as appropriate. All figures were generated using GraphPad Prism version 8.3.0 (GraphPad Software, San Diego, CA, USA).

## Figures and Tables

**Figure 1 pharmaceuticals-18-01615-f001:**
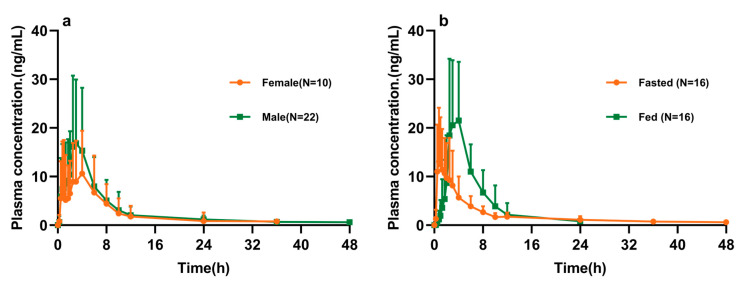
Plasma concentration–time profiles of ibrutinib in healthy Chinese subjects: (**a**) male vs. female and (**b**) fasted vs. fed states (mean ± SD). Note: Concentrations beyond 48 h were below the lower limit of quantification (LLOQ) (0.5 ng/mL).

**Figure 2 pharmaceuticals-18-01615-f002:**
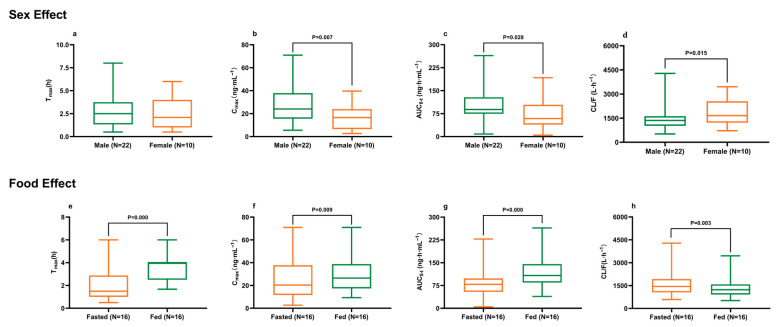
Effects of sex and food intake on Tmax (**a**,**e**), Cmax (**b**,**f**), AUC0-t (**c**,**g**), and CL/F (**d**,**h**) of ibrutinib following a single 140 mg dose in healthy Chinese subjects. *p* values were derived from the Kruskal–Wallis H test. Boxes represent the interquartile range (25th–75th percentiles); the whiskers indicate the maximum and minimum values; and the horizontal line within each box denotes the median.

**Figure 3 pharmaceuticals-18-01615-f003:**
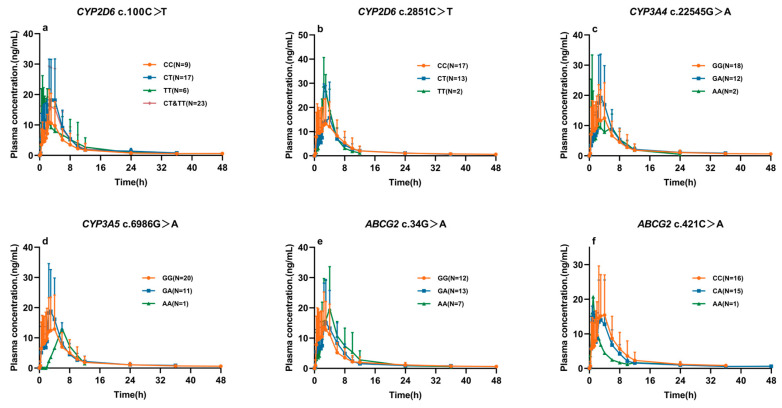
Plasma concentration–time curves of ibrutinib in healthy Chinese subjects with different genotypes: CYP2D6 c.100C>T (**a**), CYP2D6 c.2851C>T (**b**), CYP3A4 c.22545G>A (**c**), CYP3A5 c.6986G>A (**d**), ABCG2 c.34G>A (**e**), and ABCG2 c.421C>A (**f**) (mean ± SD). Note: Concentrations beyond 48 h were below the LLOQ (0.5 ng/mL). For certain genotypes (e.g., CYP2D6 CT/TT and CYP3A4 AA), plasma concentrations beyond approximately 35 h were also below the LLOQ and thus omitted from the plots.

**Figure 4 pharmaceuticals-18-01615-f004:**
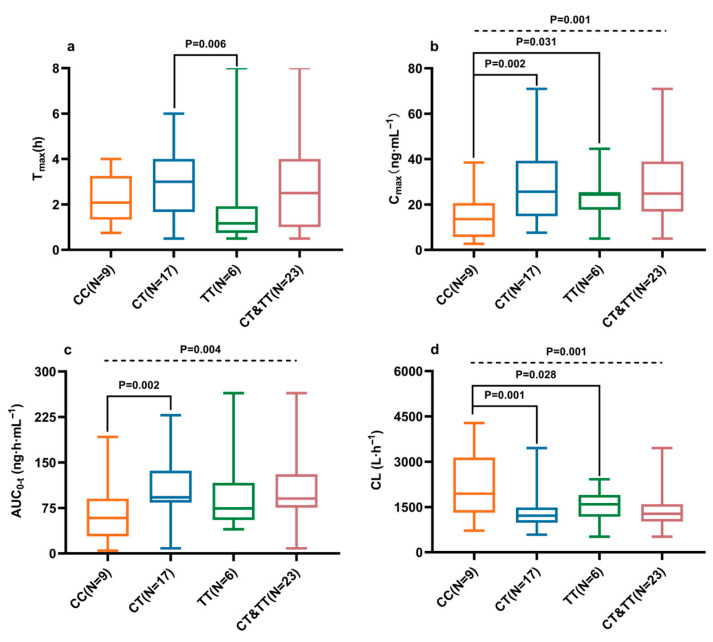
Effect of the CYP2D6 c.100C>T polymorphism on Tmax (**a**), Cmax (**b**), AUC0-t (**c**), and CL/F (**d**) of ibrutinib after a single 140 mg dose in Chinese subjects. *p* values were derived from the Mann–Whitney U test. The format of the boxes, whiskers, and medians is consistent with Figure 2.

**Figure 5 pharmaceuticals-18-01615-f005:**
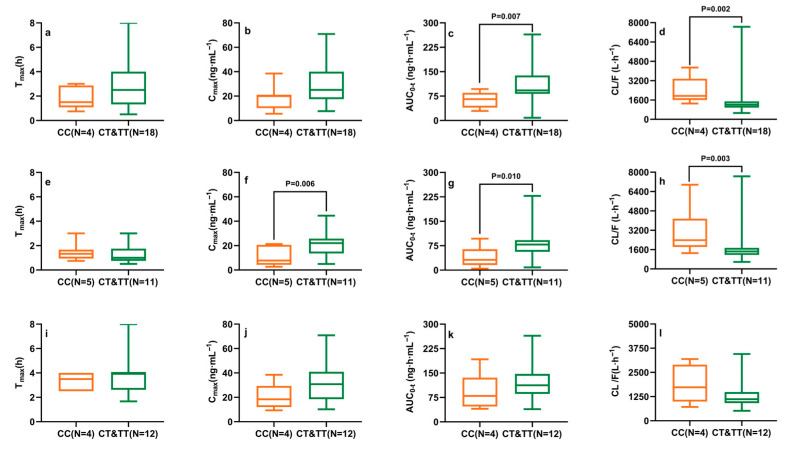
Effect of the CYP2D6 c.100C>T polymorphism on ibrutinib pharmacokinetic parameters after a single 140 mg dose in Chinese male (**a**–**d**), fasted (**e**–**h**), and fed (**i**–**l**) subgroups. *p* values were calculated using the Kruskal–Wallis H test. The formatting of the boxes, whiskers, and medians is consistent with Figure 2.

**Figure 6 pharmaceuticals-18-01615-f006:**
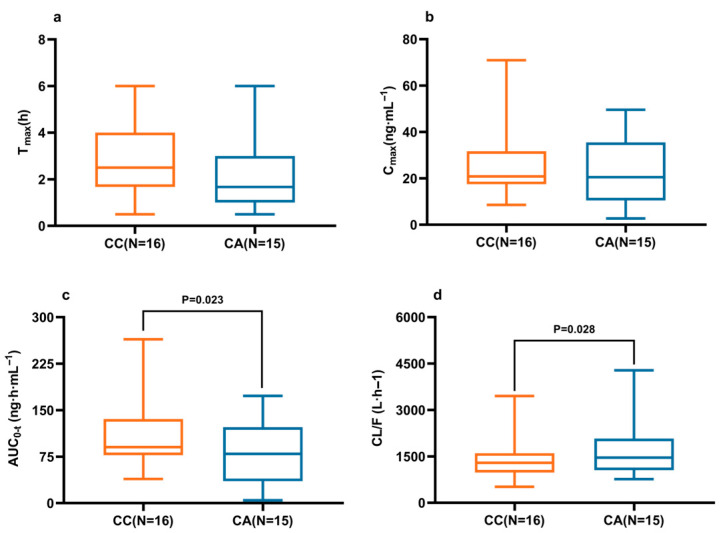
Effect of the ABCG2 c.421C>A polymorphism on Tmax (**a**), Cmax (**b**), AUC0-t (**c**), and CL/F (**d**) of ibrutinib after a single 140 mg dose in Chinese subjects. *p* values were calculated using the Kruskal–Wallis H test. The presentation of the boxes, whiskers, and medians is consistent with Figure 2.

**Table 1 pharmaceuticals-18-01615-t001:** Baseline demographics and clinical characteristics of 32 healthy Chinese subjects.

Variables	Mean (95% Confidence Interval)	Median (Quartile)
Male, N (%)	22 (68.8)	
Age (years)	33 (31, 35)	32 (29, 35)
Body Weight (kg)	69.2 (66.6, 71.8)	70.5 (62.4, 74.4)
Height (cm)	169.8 (167.3, 172.2)	170.3 (166.0, 173.9)
BMI (kg/m^2^)	24.0 (23.4, 24.5)	24.4 (22.9, 25.3)
ALT (U/L)	28 (23, 32)	26 (16, 43)
AST (U/L)	25 (22, 28)	23 (19, 29)
CREA (μmol/L)	70.5 (65.2, 75.8)	71.4 (61.2, 82.7)

**Table 2 pharmaceuticals-18-01615-t002:** Characteristics of the analyzed single-nucleotide polymorphisms.

SNP	rs ID	Genotypic Frequency N (%)			Mutant Allele	MAF ^2^				HWE ^3^
		Wild	Hetero	Variant		Study	East Asian ^1^	American ^1^	Europe ^1^	
CYP2D6 c.100C>T	rs1065852	CC, 9 (28.1)	CT, 17 (53.1)	TT, 6 (18.8)	T	0.45	0.57	0.15	0.20	0.919
CYP2D6 c.2851 C>T	rs16947	CC, 17 (53.1)	CT, 13 (40.6)	TT, 2 (6.3)	T	0.27	0.14	0.33	0.34	0.974
CYP3A4 c.22545G>A	rs2242480	GG, 18 (56.2)	GA, 12 (37.5)	AA, 2 (6.3)	A	0.25	0.27	0.39	0.08	1.000
CYP3A5 c.6986 G>A	rs776746	GG, 20 (62.5)	GA, 11 (34.4)	AA, 1 (3.1)	A	0.20	0.29	0.20	0.06	0.938
ABCG2 c.34 G>A	rs2231137	GG, 12 (37.5)	GA, 13 (40.6)	AA, 7 (21.9)	A	0.42	0.33	0.24	0.06	0.632
ABCG2 c.421 C>A	rs2231142	CC, 16 (50.0)	CA, 15 (46.88)	AA, 1 (3.13)	A	0.27	0.29	0.14	0.09	0.532

^1^ Data were obtained from the 1000 Genomes Project; ^2^ MAF, minor allele frequency; ^3^ HWE, Hardy–Weinberg equilibrium. The *p* values were calculated by the Chi-squared test.

**Table 3 pharmaceuticals-18-01615-t003:** Effects of sex and food intake on the pharmacokinetic parameters of ibrutinib in healthy Chinese subjects.

Sex	Male (N = 22)	Female (N = 10)	*p* ^1^
Tmax (h)	2.50 (1.33, 3.75)	2.09 (1.00, 4.00)	0.635
Cmax (ng·mL^−1^)	24.07 (15.60, 37.88)	16.58 (6.60, 23.92)	0.007 **
AUC0-t (ng·h·mL^−1^)	88.54 (74.56, 128.34)	58.91 (39.15, 103.80)	0.028 *
CL/F (L·h^−1^)	1413 (1031, 1702)	2021 (1242, 3177)	0.015 *
**Food**	**Fasted (N = 16)**	**Fed (N = 1** **6** **)**	***p* ^1^**
Tmax (h)	1.33 (0.75, 1.67)	4.00 (2.50, 4.00)	0.000 ***
Cmax (ng·mL^−1^)	18.68 (8.68, 24.88)	26.51 (17.44, 38.77)	0.009 **
AUC0-t (ng·h·mL^−1^)	73.56 (35.45, 88.52)	107.73 (84.76, 145.50)	0.000 ***
CL/F (L·h^−1^)	1692 (1313, 2615)	1225 (913, 1581)	0.003 **

^1^ Data were presented as median (the interquartile range). * *p* < 0.05, ** *p* < 0.01, *** *p* < 0.001.

**Table 4 pharmaceuticals-18-01615-t004:** Pharmacokinetic parameters of ibrutinib following oral administration in subjects carrying CYP2D6 c.100C>T, CYP2D6 c.2851C>T, CYP3A4 c.22545 G>A, CYP3A5 c.6986 G>A, ABCG2 c.421C>A, and ABCG2 c.34G>A genotypes.

CYP2D6 c.100C>T	CC (N = 9)	CT (N = 17)	TT (N = 6)	CT&TT (N = 23)
Tmax (h)	2.09 (1.33, 3.25)	3.00 (1.67, 4.00)	1.17 (0.75, 1.92)	2.50 (1.00, 4.00)
Cmax (ng·mL^−1^)	13.59 (5.71, 20.61)	25.66 (14.88, 39.27)	24.4 (17.73, 25.48)	24.84 (16.89, 38.93)
AUC0-t (ng·h·mL^−1^)	58.54 (28.21, 90.50)	92.49 (83.83, 136.53)	74.15 (55.13, 116.32)	90.47 (75.45, 130.79)
CL/F (L·h^−1^)	2124 (1481, 3910)	1222 (1003, 1560)	1591 (1178, 1900)	1356 (1021, 1618)
**CYP2D6 c.2851 C**>**T**	**CC (N = 17)**	**CT (N = 13)**	**TT (N = 2)**	** *p* **
Tmax (h)	1.67 (1.00, 3.25)	3.00 (1.33, 4.00)	2.50 (2.50, 2.88)	0.400
Cmax (ng·mL^−1^)	22.53 (16.89, 32.87)	18.43 (10.47, 34.81)	26 (16.68, 37.00)	0.445
AUC0-t (ng·h·mL^−1^)	86.12 (60.90, 115.65)	84.48 (38.53, 131.77)	79.46 (63.92, 131.59)	0.744
CL/F (L·h^−1^)	1468 (1133, 1729)	1293 (1003, 3152)	1731 (1059, 2113)	0.889
**CYP3A4 c.22545 G**>**A**	**GG (N = 18)**	**GA (N = 12)**	**AA (N = 2)**	***p* ^1^**
Tmax (h)	4.25 (1.19, 6.00)	3.00 (1.75, 4.00)	1.67 (1.00, 3.75)	0.077
Cmax (ng·mL^−1^)	18.92 (12.81, 39.40)	22.03 (15.57, 35.64)	20.40 (10.38, 28.48)	0.531
AUC0-t (ng·h·mL^−1^)	78.72 (51.74, 114.23)	92.27 (81.83, 133.09)	74.89 (39.19, 107.13)	0.075
CL/F (L·h^−1^)	1663 (1208, 2693)	1249 (1016, 1524)	1653 (1110, 3177)	0.033 *
**CYP3A5 c.6986 G**>**A**	**GG (N = 20)**	**GA (N = 11)**	**AA (N = 1)**	***p* ^2^**
Tmax (h)	1.67 (1.00, 4.00)	2.50 (1.59, 3.25)	--	0.038 *
Cmax (ng·mL^−1^)	20.40 (11.00, 31.58)	24.07 (16.80, 36.64)	13.15	0.251
AUC0-t (ng·h·mL^−1^)	82.92 (40.62, 121.79)	92.27 (75.70, 129.08)	64.67	0.134
CL/F (L·h^−1^)	1559 (1078, 3024)	1249 (990, 1618)	2273.82	0.066
**ABCG2 c.34G**>**A**	**GG (N = 12)**	**GA (N = 13)**	**AA (N = 7)**	** *p* **
Tmax (h)	1.84 (1.08, 3.00)	2.50 (1.00, 4.00)	2.75 (1.92, 4.00)	0.278
Cmax (ng·mL^−1^)	19.26 (9.43, 35.36)	21.18 (16.61, 32.87)	24.4 (13.54, 36.09)	0.502
AUC0-t (ng·h·mL^−1^)	83.9 (60.80, 103.8)	85.48 (54.43, 112.27)	98.18 (56.02, 152.79)	0.676
CL/F (L·h^−1^)	1531 (1091, 1936)	1452 (1104, 2159)	1347 (867, 2302)	0.744
**ABCG2 c.421C**>**A**	**CC (N = 16)**	**CA (N = 15)**	**AA (N = 1)**	***p* ^3^**
Tmax (h)	2.50 (1.67, 4.00)	1.67 (1.00, 3.00)	1.34	0.158
Cmax (ng·mL^−1^)	20.81 (17.44, 31.71)	20.48 (10.47, 35.49)	7.04	0.095
AUC0-t (ng·h·mL^−1^)	90.47 (77.28, 135.64)	74.89 (32.78, 116.06)	50.33	0.023 *
CL/F (L·h^−1^)	1293 (980, 1602)	1631 (1129, 3339)	1879	0.028 *

^1^ Data were presented as median (interquartile range). * *p* < 0.05; ^2^ *p* values indicate differences in pharmacokinetic parameters of ibrutinib between GG and GA genotypes; ^3^ *p* values indicate differences in pharmacokinetic parameters of ibrutinib between CC and CA genotypes.

**Table 5 pharmaceuticals-18-01615-t005:** Primer sequences for genotyping polymorphic CYP2D6, CYP3A4/5, and ABCG2 genes.

SNP (rs ID)	Primer Name	Primer Sequence	Sequencing Sequence	Sequencing Primer	Length (bp)
CYP2D6 c.100C>T	CYP2D6-rs1065852-F	TGCCATGTATA AATGCCCTTCT	CYP2D6-rs1065852-RS	TGAGGCAGGT ATGGGGCTAG	355
(rs1065852)	CYP2D6-rs1065852-R	TTGGTAGTGAG GCAGGTATGG			
CYP2D6 c.2851 C>T	CYP2D6-rs16947-F	CGGGTGTCCC AGCAAAGTT	CYP2D6-rs16947-FS	CCTCGGCCCC TGCACTGTT	264
(rs16947)	CYP2D6-rs16947-R	CTACCCCGTT CTGTCCCGA			
CYP3A4 c.22545G>A	CYP3A4-1G-F	AGAGCCTTCCTA CATAGAGTCAG	CYP3A4-1G-S	GCAGTGTTCTCT CCTTCATTATG	394
(rs2242480)	CYP3A4-1G-R	CCTTAGGGAT TTGAGGGCTT			
CYP3A5 c.6986A>G	CYP3A5-2-F	TAGTAGACAGA TGACACAGCTC	CYP3A5-S	GAGAGTGGCAT AGGAGATACC	569
(rs776746)	CYP3A5-2-R	TCACTAGCACT GTTCTGATCAC			
ABCG2 c.34G>A	rs2231137-F	TGCCTGTCTTC CCATTTAGGTT	rs2231137-F	TGCCTGTCTTC CCATTTAGGTT	474
(rs2231137)	rs2231137-R	AGCCAAAACCTG TGAGGTTCACTG			
ABCG2 c.421G>T	rs2231142-F	AAACAGTCATGGT CTTAGAAAAGAC	rs2231142-F	AAACAGTCATGGT CTTAGAAAAGAC	197
(rs2231142)	rs2231142-R	AGACCTAACTCT TGAATGACCCT			

## Data Availability

The original contributions presented in this study are included in the article/Appendix A. Further inquiries can be directed to the corresponding authors.

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
