# Peer review of "Influence of CYP2D6, CYP3A, and ABCG2 Genetic Polymorphisms on Ibrutinib Disposition in Chinese Healthy Subjects"

_pharmaceuticals, 2025, doi:10.3390/ph18111615_

Round 1
Reviewer 1 Report
Comments and Suggestions for Authors
Here are a few suggestions for authors to improve the manuscript
- The UPLCMS/MS method was a new method validated by the authors, or it is an already reported method. If it is already a reported method, then provide a reference. If it is a new method, then more detail is required, including chromatograms, accuracy, and precision data that need to be provided. Was an internal standard used?
- The fed and fasted states and gender (Male and female) studies were conducted in different phases, or they were separated from a single study. It is not clear whether the gender studies are conducted in a fasted or fasted state.
- In section 4, it is mentioned that the samples were collected up to 60 hours, but in figures 1 and 3, the scale is shown up to 48 hours.
- In Figure 3a, the CT is quantified up to 35 hours; why not beyond 35 hours? Same query about TT, in Figure 3B, AA in Figure 3C. Such a pattern is also seen in other figures. Please justify it.
- Figures 4 and 5, captions are not properly placed.
- In the discussion, the comparison between the Chinese population and other populations needs to be stated. How much difference is there?
- Similarly, in the fed state, any difference was found in the previously reported study and the current study?
Comments on the Quality of English Language
The English language can be improved.
Author Response
- The UPLCMS/MS method was a new method validated by the authors, or it is an already reported method. If it is already a reported method, then provide a reference. If it is a new method, then more detail is required, including chromatograms, accuracy, and precision data that need to be provided. Was an internal standard used?
Response: We thank the reviewer for the valuable comment. The UPLC–MS/MS method for the determination of ibrutinib in plasma was newly developed and validated in our laboratory according to ICH M10 (2022) bioanalytical method validation guideline. To clarify, we have now specified that ibrutinib-d₅ was used as the internal standard, and we briefly summarized the validation characteristics (linearity, accuracy, and precision) in Section 4.2 of the revised manuscript. Representative chromatograms are provided in the Supplementary Information (Figure S1). Detailed validation data will be included in a separate methodological paper currently in preparation.
- The fed and fasted states and gender (Male and female) studies were conducted in different phases, or they were separated from a single study. It is not clear whether the gender studies are conducted in a fasted or fasted state.
Response: We appreciate the reviewer’s insightful question. The current analysis was based on a randomized two-period crossover design in which each participant received ibrutinib under both fasting and fed conditions with an adequate washout interval. Therefore, the effects of food were assessed within subjects, minimizing interindividual variability. Analyses of gender and genotype effects were exploratory and performed using data from both conditions. To clarify this point, we have revised the “Study Design” section accordingly. In addition, we have noted this design feature as a limitation in the Discussion, acknowledging that although within-subject comparison reduces variability, residual confounding by sex or genotype between conditions cannot be completely excluded.
- In section 4, it is mentioned that the samples were collected up to 60 hours, but in figures 1 and 3, the scale is shown up to 48 hours.
Response: Thank you for your careful observation. In this study, blood samples were indeed collected up to 60 hours post-dose to ensure full pharmacokinetic profiling and accurate estimation of the terminal elimination phase. However, for better visualization and to focus on the main absorption and distribution phases, the concentration–time profiles in Figures 1 and 3 were truncated at 48 hours, as plasma concentrations beyond this time point were below the lower limit of quantification. We have clarified this in the figure legends of the revised manuscript.
- In Figure 3a, the CT is quantified up to 35 hours; why not beyond 35 hours? Same query about TT, in Figure 3B, AA in Figure 3C. Such a pattern is also seen in other figures. Please justify it.
Response: We sincerely thank the reviewer for this valuable comment. The apparent truncation of some genotype curves in Figure 3 was due to plasma concentrations at later time points falling below the lower limit of quantification (LLOQ, 0.5 ng/mL) in most participants carrying these alleles. Consequently, the mean concentration values could not be reliably calculated beyond approximately 35 hours for certain genotypes (e.g., CYP2D6 CT, TT, and CYP3A4 AA). To ensure data accuracy and avoid misleading extrapolation, we displayed only the quantifiable concentration range for each genotype. This clarification has now been added to the figure legend.
- Figures 4 and 5, captions are not properly placed.
Response: We thank the reviewer for the careful observation. We have corrected the figure layout and caption placement according to the journal’s formatting requirements. In the previous version, the image intended for Figure 4 was mistakenly replaced by a duplicate of Figure 5, resulting in two identical figures. This error has now been corrected in the revised manuscript: the appropriate Figure 4 has been reinserted, and all figure numbering, captions, and in-text citations have been synchronized accordingly. Additionally, the captions of Figures 4 and 5 have been refined for clarity and consistency.
- In the discussion, the comparison between the Chinese population and other populations needs to be stated. How much difference is there?
Response: Thank you for this insightful suggestion. We have added a concise paragraph in the Discussion (highlighted in yellow) addressing interethnic pharmacogenetic differences. Using 1000 Genomes data, we note that CYP2D6 c.100C>T and ABCG2 c.421C>A are notably more common in East Asians than in Europeans or Americans, indicating that genotype–PK associations in our Chinese cohort may not apply uniformly across populations. We also briefly reference prior evidence linking reduced-function CYP2D6 alleles and ABCG2 c.421A to altered drug exposure and toxicity, which is consistent with our findings. In addition, we cite a recent pharmacogenetic study reporting that variants in genes such as KCNQ1, GATA4, CYP2D6, CYP3A4/5, and ABCB1 were associated with cardiovascular adverse events during ibrutinib therapy. Although that work focused on clinical outcomes rather than pharmacokinetics, it supports the broader relevance of pharmacogenomic variability. These additions strengthen the contextualization of our results and highlight the need to validate genotype–PK associations in diverse populations. The revisions appear in the Discussion and are marked in yellow.
- Similarly, in the fed state, any difference was found in the previously reported study and the current study?
Response: Thank you for this valuable suggestion. We agree that a clearer comparison with previously reported food-effect studies is necessary. In the revised Discussion section, we have added a direct comparison between our findings and both international and Chinese data. Specifically, our study showed that systemic exposure to ibrutinib increased by approximately 1.7-fold under fed conditions compared with fasting. This result is consistent with prior evidence. The U.S. FDA prescribing information reports that administration of ibrutinib with a high-fat, high-calorie meal (800–1,000 kcal, ~50% fat) increases Cmax by 2–4 fold and AUC by around twofold. Similarly, de Jong et al. (2015) observed a 2–3-fold rise in Cmax and nearly a doubling of AUC in non-Chinese volunteers, while a recent four-period crossover study in healthy Chinese adults reported a 2.2-fold increase in Cmax and a 1.3-fold increase in AUC. Therefore, our data fall within the expected range across populations and study designs. The revised paragraph has been added to the Discussion.
Reviewer 2 Report
Comments and Suggestions for Authors
This work addresses a clinically relevant pharmacogenetic question and presents potentially novel findings. The manuscript investigates how dietary state, gender, and selected pharmacogenetic variants in CYP2D6, CYP3A4, CYP3A5, and ABCG2 influence the pharmacokinetics of ibrutinib in 32 healthy Chinese adults after a single 140 mg dose. The study is promising, but publication should await resolution of the discrepancies. The small cohort size, lack of multiple-testing correction, and limited discussion on certain contradictory findings temper the impact. In addition, several critical methodological, reporting, and internal-consistency issues must be resolved. The topic is clinically relevant and the dataset is potentially publishable after substantial revision.
Here are my detailed comments and suggestions:
- Methods state 32 participants randomized equally to fasting (n=16) and fed (n=16), yet Table 3 analyzes fed N=10 (fasted N=16). No explanation of attrition/exclusion is provided.
- Add PK parameters AUC0–∞, λz, t1/2, Vz/F, % extrapolated AUC, and noncompartmental analysis settings. Replace CL/F derived from AUC0–t or clearly state if this was used.
- The Discussion cites a reduction of apparent volume of distribution from 3466 L (fasting) to 1347 L (fed), but Vz/F is not presented in Tables/Figures nor described in Methods.
- The assay description lists instrument, transitions, and range but lacks validation data like selectivity, accuracy/precision at LLOQ/low/medium/high, carryover, matrix effect, recovery, dilution integrity, stability, incurred sample reanalysis.
- “High-fat meal 30 min prior” is vague. Provide the menu/energy/fat content and timing relative to dosing per regulatory guidance.
- Numerous hypothesis tests are conducted with no correction for multiple comparisons, raising type I error concerns.
- Implement a pre-specified analysis plan, adjust for multiplicity, and report effect sizes with 95% CIs.
- Sex is imbalanced (22 M/10 F). It is unclear whether genotype frequencies are balanced across fed/fasted and sex strata.
- Define genotype-to-phenotype groupings (e.g., CYP2D6 metabolizer categories) a priori. Clarify whether haplotypes (e.g., CYP3A41G/CYP3A53) were considered.
- State whether sample analysis was blinded to genotype/feeding group; include reference to raw data/code availability if possible.
- Describe allocation method, any analyst blinding, and handling of outliers/missing samples.
- No methods/source for the solubility experiment are provided. Either add experimental details or cite the source clearly.
- Temper conclusions around ABCG2 421A (counter-intuitive direction) until adjusted analyses and multiplicity control are shown.
- Clearly state that results are preliminary and need confirmation in patient populations under therapeutic conditions.
- Correct figure captions/labels (e.g., Figure 4), ensure panel lettering matches text, and document any excluded subjects per figure. Provide details on methods/source for Figure 7.
Author Response
- Methods state 32 participants randomized equally to fasting (n=16) and fed (n=16), yet Table 3 analyzes fed N=10 (fasted N=16). No explanation of attrition/exclusion is provided.
Response: We thank the reviewer for identifying this inconsistency. This was a typographical error in Table 3: the fed group sample size was incorrectly indicated as n = 10 instead of the correct n = 16. All 32 randomized participants completed the study, and no participants were excluded. We have corrected Table 3 to indicate fed (n = 16), and these corrections are highlighted in yellow in the revised manuscript. We apologize for the oversight and any confusion it caused.
- Add PK parameters AUC0–∞, λz, t1/2, Vz/F, % extrapolated AUC, and noncompartmental analysis settings. Replace CL/F derived from AUC0–t or clearly state if this was used.
Response: Thank you for this constructive suggestion. We confirm that CL/F in the original analysis was calculated using AUC0-∞ rather than AUC0-t. To avoid ambiguity and to address the reviewer’s request, we have revised the Pharmacokinetic Analysis section to provide the full details of the noncompartmental analysis and to include the additional PK parameters. The updated text reads as follows: Noncompartmental analysis was performed using Phoenix WinNonlin version 8.2 (Certara, USA). The following pharmacokinetic parameters were determined: Cmax, Tmax, AUC0-t, CL/F, AUC0-∞, λz, t₁/₂, Vz/F, and % extrapolated AUC. λz was derived from the log-linear terminal phase of the plasma concentration–time curve, with t₁/₂ calculated as ln (2)/λz. AUC0–∞ was computed as AUC0–t plus C_last/λz. Apparent clearance (CL/F) was calculated as Dose/AUC0–∞, and apparent volume of distribution (Vz/F) was determined as (CL/F)/λz. The percentage of extrapolated AUC was calculated as (AUC0-∞ − AUC0-t)/AUC0-∞ × 100%. In accordance with the comment, we have now included additional pharmacokinetic parameters (AUC0-∞, λz, t₁/₂, Vz/F, and % extrapolated AUC) in the revised manuscript. These data have been compiled in Supplementary Table S1 to ensure clarity without expanding the main text.
- The Discussion cites a reduction of apparent volume of distribution from 3466 L (fasting) to 1347 L (fed), but Vz/F is not presented in Tables/Figures nor described in Methods.
Response: We thank the reviewer for this insightful comment. We acknowledge that the median apparent volume of distribution (Vz/F) cited in the Discussion was incorrect. The correct values should be a reduction from 18727 L (fasting) to 3984 L (fed), rather than 3466 L to 1347 L. We have corrected this in the revised Discussion. To address the reviewer’s concern, Vz/F has now been added as an additional pharmacokinetic parameter in Supplementary Table S1, with data presented separately for the fasting and fed groups.
- The assay description lists instrument, transitions, and range but lacks validation data like selectivity, accuracy/precision at LLOQ/low/medium/high, carryover, matrix effect, recovery, dilution integrity, stability, incurred sample reanalysis.
Response: We appreciate both reviewers’ insightful comments regarding the bioanalytical method. The UPLC–MS/MS method for the quantification of ibrutinib in human plasma was developed and validated in our laboratory according to the ICH M10 (2022) bioanalytical method validation guidelines. The assay employed ibrutinib-d₅ as an internal standard. Validation parameters—including selectivity, linearity, accuracy, precision, recovery, matrix effect, carryover, stability, and incurred sample reanalysis—all met regulatory acceptance criteria (±15%, ±20% for LLOQ). Representative chromatograms are provided in the Supplementary Information (Figure S1). Detailed validation data will be presented in a separate methodological paper currently in preparation. Corresponding clarifications have been added to Section 4.2 of the revised manuscript.
- “High-fat meal 30 min prior” is vague. Provide the menu/energy/fat content and timing relative to dosing per regulatory guidance.
Response: Thank you for pointing this out. We have added a detailed description of the high-fat meal in the Methods section. Specifically, the high-fat, high-calorie meal was provided approximately 30 minutes before ibrutinib administration, in accordance with China NMPA and US FDA bioavailability/bioequivalence study guidance. The meal contained approximately 800–1000 kcal, with about 50% of total calories from fat (≈500 kcal), 35% from carbohydrates, and 15% from protein. The menu typically consisted of fried eggs, whole milk, buttered bread, and fried potatoes.
- Numerous hypothesis tests are conducted with no correction for multiple comparisons, raising type I error concerns.
Response: We appreciate the reviewer’s concern regarding the potential inflation of type I error due to multiple comparisons. In this study, the primary aim was to explore potential pharmacogenetic associations rather than to establish confirmatory conclusions; therefore, the analyses were considered exploratory in nature. Nevertheless, to improve statistical rigor, we have now acknowledged this limitation in the Discussion section.
- Implement a pre-specified analysis plan, adjust for multiplicity, and report effect sizes with 95% CIs.
Response: We are grateful for this constructive suggestion. As this work was conceived as an exploratory study, the analyses were not designed under a pre-specified statistical analysis plan, and adjustments for multiplicity or effect-size estimation with confidence intervals were not incorporated at the outset. We fully acknowledge that such approaches would enhance the robustness and interpretability of the findings. To avoid overinterpretation, we have now emphasized in the Discussion that the results should be viewed as hypothesis-generating and interpreted with caution. We agree that future confirmatory studies with larger sample sizes will benefit from predefined analytical frameworks, multiplicity control, and effect-size reporting.
- Sex is imbalanced (22 M/10 F). It is unclear whether genotype frequencies are balanced across fed/fasted and sex strata.
Response: We appreciate the reviewer’s insightful observation. Although genotypes were not stratified by sex or dietary status in the statistical analysis, no intentional imbalance was introduced during randomization. However, given the limited sample size and unequal sex ratio, residual confounding by genotype–sex or genotype–diet interactions cannot be fully ruled out. To address this concern, we have now acknowledged this issue as a study limitation in the revised Discussion section.
- Define genotype-to-phenotype groupings (e.g., CYP2D6 metabolizer categories) a priori. Clarify whether haplotypes (e.g., CYP3A41G/CYP3A53) were considered.
Response: We thank the reviewer for this insightful comment. In this exploratory pharmacokinetic study, genotype–phenotype groupings (such as CYP2D6 metabolizer categories) were not predefined. Instead, single-locus analyses were conducted based on individual SNPs to identify potential associations with pharmacokinetic variability. Haplotype-based analyses (e.g., CYP3A4*1G/CYP3A5*3 combinations) were not performed due to the limited sample size, which restricted statistical power for multi-locus stratification. To address this point more clearly, we have added explanatory text to the Discussion section in the revised manuscript.
- State whether sample analysis was blinded to genotype/feeding group; include reference to raw data/code availability if possible.
Response: Thank you for raising this point. The quantification of ibrutinib concentrations was performed using a validated and standardized UPLC–MS/MS protocol, in which measurements are instrument-based and not subject to operator interpretation. The analysts performing the assays were not involved in either genotyping or feeding group assignment, which further reduces any theoretical risk of bias. While the raw datasets and code are not publicly archived due to institutional restrictions, they can be provided by the corresponding author upon reasonable request. A clarifying statement has been added to the Methods section.
- Describe allocation method, any analyst blinding, and handling of outliers/missing samples.
Response: Thank you for this helpful suggestion. In this study, participants were randomly assigned to the fasting or fed phase using a simple randomization approach without stratification by sex or genotype. Plasma drug concentrations were quantified through automated UPLC–MS/MS procedures, and the analytical personnel did not participate in sample allocation or genotyping. No samples were excluded as outliers, and there were no missing pharmacokinetic data points for the analyzed subjects. To improve transparency, we have now added brief descriptions of the allocation process and sample handling in the revised Methods section.
- No methods/source for the solubility experiment are provided. Either add experimental details or cite the source clearly.
Response: Thank you for pointing this out. The solubility figure previously included in the manuscript (Figure 7) was based on internal, unpublished laboratory data and has now been removed to avoid any ambiguity regarding source or methodology. Instead, we supplemented the relevant section of the Discussion with published data from the U.S. FDA chemistry review of ibrutinib (NDA 210563), which reports that ibrutinib is practically insoluble to slightly soluble in aqueous media across physiological pH ranges. Specifically, the FDA document indicates that ibrutinib solubility is approximately 1.6 mg/mL at pH 1 but decreases to 0.003 mg/mL at pH 4.5 and remains practically insoluble in the pH range of 3–8. These reference data have been cited in the revised text to support the discussion on pH-dependent solubility. No additional solubility experiment is presented in the revised manuscript.
- Temper conclusions around ABCG2 421A (counter-intuitive direction) until adjusted analyses and multiplicity control are shown.
Response: Thank you for this valuable comment. We agree that the observed association between the ABCG2 c.421A allele and increased ibrutinib clearance appears counterintuitive compared with previous findings on BCRP function. As suggested, we have tempered the corresponding conclusions and clarified that this finding should be interpreted cautiously until confirmed by adjusted analyses and larger studies with multiplicity control. Relevant revisions have been made in the Discussion section.
- Clearly state that results are preliminary and need confirmation in patient populations under therapeutic conditions.
Response: We thank the reviewer for this valuable suggestion. We agree that the current findings are preliminary and derived from healthy volunteers rather than patients receiving therapeutic dosing. To address this point, we have now added a statement in the Discussion acknowledging that confirmation in patient populations under clinical treatment conditions is needed to establish the translational relevance of these results.
- Correct figure captions/labels (e.g., Figure 4), ensure panel lettering matches text, and document any excluded subjects per figure. Provide details on methods/source for Figure 7.
Response: Thank you for pointing this out. We carefully reviewed the figures and confirmed that this was a formatting error during figure placement. The original Figure 4 was mistakenly replaced by a duplicate of Figure 5, and the panel label “b” in Figure 4 was incorrectly marked as “d”. All figures have been corrected and verified to match the corresponding captions and text. Besides, all figures include data from the full cohort of 32 participants. No subjects were excluded from any of the pharmacokinetic analyses or plots. Some concentration–time curves do not show data points beyond 48 hours only because plasma concentrations were below the LLOQ, but these participants were not removed from the analysis. The previously submitted Figure 7 was derived from internal laboratory data on the saturated solubility of ibrutinib and had not been formally published, which could lead to ambiguity. To ensure transparency and compliance with publication standards, we have taken the following action: Figure 7 has been removed from the revised manuscript. Its content is now described in text form and supported by citation of the FDA regulatory document (as referenced above).
Round 2
Reviewer 1 Report
Comments and Suggestions for Authors
The authors have improved the manuscript. The main manuscript is fine, but the chromatograms in the supplementary figures need further clarification. The authors must describe what the blue and red lines (chromatograms) represent. Could you also label the peaks? The internal standard peak is missing.
Author Response
The authors have improved the manuscript. The main manuscript is fine, but the chromatograms in the supplementary figures need further clarification. The authors must describe what the blue and red lines (chromatograms) represent. Could you also label the peaks? The internal standard peak is missing.
Response: Thank you for your constructive suggestion. We have revised the Supplementary Figure S1 to clearly indicate the identities of the chromatographic peaks and the corresponding traces.
Line colors clarified in the figure legend: The blue trace represents the MRM transition for ibrutinib (m/z 441.3 → 138.1). The red trace represents the MRM transition for the internal standard (ibrutinib-d₅) (m/z 446.3 → 309.1).
Peak annotations added directly on the chromatograms: The retention times of both ibrutinib and the internal standard are now labeled.
Reviewer 2 Report
Comments and Suggestions for Authors
The authors have substantially improved their manuscript. Most of the previously raised concerns have been carefully addressed through thoughtful clarifications, additional data inclusion, and appropriate methodological corrections. The manuscript is now suitable for publication, based on editorial clarifications.
Author Response
The authors have substantially improved their manuscript. Most of the previously raised concerns have been carefully addressed through thoughtful clarifications, additional data inclusion, and appropriate methodological corrections. The manuscript is now suitable for publication, based on editorial clarifications.
Response: We sincerely appreciate the reviewer’s positive assessment and recognition of the substantial improvements made in the revised manuscript. We are grateful that the manuscript is now considered suitable for publication pending editorial clarifications. Thank you again for your time and constructive feedback throughout the review process.